# The lymphatic filariasis treatment study landscape: A systematic review of study characteristics and the case for an individual participant data platform

**Luzia T. Freitas**[1,2,3⊙], **Mashroor Ahmad Khan**[4⊙], **Azhar Uddin**[4⊙], **Julia B. Halder**[1,2,3,5], **Sauman Singh-Phulgenda**[3,6], **Jeyapal Dinesh Raja**[4], **Vijayakumar Balakrishnan**[4], **Eli Harriss**[7], **Manju Rahi**[4], **Matthew Brack**[3,6], **Philippe J. Guérin**[3,6], **Maria-Gloria Basáñez**[1,2,3], **Ashwani Kumar**[8], **Martin Walker**[2,3,5⊙]*, **Adinarayanan Srividya**[4⊙]*

**1** MRC Centre for Global Infectious Disease Analysis, Department of Infectious Disease Epidemiology, School of Public Health, Imperial College London, London, United Kingdom, **2** London Centre for Neglected Tropical Disease Research, Department of Infectious Disease Epidemiology, School of Public Health, Imperial College London, London, United Kingdom, **3** Infectious Diseases Data Observatory, University of Oxford, Oxford, United Kingdom, **4** ICMR-Vector Control Research Centre, Puducherry, India, **5** Department of Pathobiology and Population Sciences, Royal Veterinary College, Hatfield, United Kingdom, **6** Centre for Tropical Medicine and Global Health, Nuffield Department of Medicine, University of Oxford, Oxford, United Kingdom, **7** The Knowledge Centre, Bodleian Health Care Libraries, University of Oxford, Oxford, United Kingdom, **8** Saveetha Institute of Medical and Technical Sciences, Saveetha University, Chennai, India

⊙ These authors contributed equally to this work.
* mwalker@rvc.ac.uk (MW); vidyaadi@gmail.com (AS)

**Data Availability Statement:** All relevant data are within the manuscript and its Supporting information files.

## Abstract

### Background

Lymphatic filariasis (LF) is a neglected tropical disease (NTD) targeted by the World Health Organization for elimination as a public health problem (EPHP). Since 2000, more than 9 billion treatments of antifilarial medicines have been distributed through mass drug administration (MDA) programmes in 72 endemic countries and 17 countries have reached EPHP. Yet in 2021, nearly 900 million people still required MDA with combinations of albendazole, diethylcarbamazine and/or ivermectin. Despite the reliance on these drugs, there remain gaps in understanding of variation in responses to treatment. As demonstrated for other infectious diseases, some urgent questions could be addressed by conducting individual participant data (IPD) meta-analyses. Here, we present the results of a systematic literature review to estimate the abundance of IPD on pre- and post-intervention indicators of infection and/or morbidity and assess the feasibility of building a global data repository.

### Methodology

We searched literature published between 1st January 2000 and 5th May 2023 in 15 databases to identify prospective studies assessing LF treatment and/or morbidity management and disease prevention (MMDP) approaches. We considered only studies where individual

**Funding:** We acknowledge funding from the Bill & Melinda Gates Foundation (Grant number: INV-004713). LTF, JBH and MGB also acknowledge funding from the MRC Centre for Global Infectious Disease Analysis (MR/R015600/1), jointly funded by the UK Medical Research Council (MRC) and the UK Foreign, Commonwealth & Development Office (FCDO), under the MRC/FCDO Concordat agreement and is also part of the EDCTP2 programme supported by the European Union. The funders had no role in study design, data collection and analysis, decision to publish, or preparation of the manuscript.

**Competing interests:** The authors have declared that no competing interests exist.

participants were diagnosed with LF infection or disease and were followed up on at least one occasion after receiving an intervention/treatment.

## Principal findings

We identified 138 eligible studies from 23 countries, having followed up an estimated 29,842 participants after intervention. We estimate 14,800 (49.6%) IPD on pre- and post-intervention infection indicators including microfilaraemia, circulating filarial antigen and/or ultrasound indicators measured before and after intervention using 8 drugs administered in various combinations. We identified 33 studies on MMDP, estimating 6,102 (20.4%) IPD on pre- and post-intervention clinical morbidity indicators only. A further 8,940 IPD cover a mixture of infection and morbidity outcomes measured with other diagnostics, from participants followed for adverse event outcomes only or recruited after initial intervention.

## Conclusions

The LF treatment study landscape is heterogeneous, but the abundance of studies and related IPD suggest that establishing a global data repository to facilitate IPD meta-analyses would be feasible and useful to address unresolved questions on variation in treatment outcomes across geographies, demographics and in underrepresented groups. New studies using more standardized approaches should be initiated to address the scarcity and inconsistency of data on morbidity management.

## Author summary

Lymphatic filariasis (LF) is a debilitating parasitic disease that the World Health Organization (WHO) has earmarked for elimination by 2030 through a combination of mass distribution of anti-parasitic medicines and disease management approaches. Great strides have been made towards the elimination of LF as a public health problem, but nearly 900 million people still require treatment every year. In recent years, new combinations of medicines have been shown to improve the treatment of LF, yet there remain substantial gaps in understanding of apparent variability in treatment success and on the best treatment and management approaches to alleviate chronic morbidity. Some of these questions could be addressed through the development of an LF global data platform, which would enable pooled analyses of all available individual participant data. Here, we present the results of a systematic literature review of the LF treatment study landscape. We estimate the abundance of individual data on treatment of infection and morbidity and describe the characteristics of the studies and participants that have generated these data. We argue that collating and curating these data into a data LF platform could help to fill gaps in understanding of the best ways to treat infection and disease and enhance prospects of eliminating LF by 2030.

## Introduction

Lymphatic filariasis (LF) is a mosquito-borne neglected tropical disease (NTD) targeted by the World Health Organization (WHO) for elimination as a public health problem (EPHP), aiming at circa 80% of endemic countries validated for EPHP by 2030 [1]. Globally, approximately 50 million people are infected with the filarial nematodes (*Wuchereria bancrofti*, *Brugia malayi*

and *B. timori)* that cause LF, and 885 million people are estimated to be at risk of infection [2]. Moreover, approximately 36 million people are chronically debilitated by filarial lymphedema [3]. The global public health burden of LF in 2019 was estimated as 1.63 million disability-adjusted life years (DALYs) [4].

Since the inception of the Global Programme to Eliminate Lymphatic Filariasis (GPELF) in 2000, 17 countries have eliminated LF as a public health problem, primarily by repeated rounds of mass drug administration (MDA) with anti-filarial drugs to at-risk populations, a strategy known as preventive chemotherapy. The combination therapies of diethylcarbamazine plus albendazole (DA), or ivermectin plus albendazole (IA, used in Africa where co-endemicity with other filarial diseases, particularly onchocerciasis, complicates the use of diethylcarbamazine), were the cornerstones of MDA between 2000 and 2019, with more than 9 billion treatments distributed among endemic countries [2]. Since 2019, the use of a triple drug regimen combining ivermectin, diethylcarbamazine and albendazole (IDA), which has demonstrated superior efficacy compared to its dual therapy counterparts, (i.e., DA or IA) [5–8], has been increasingly adopted to accelerate progress towards LF elimination [9,10].

In addition to MDA, the LF elimination strategy also consists of morbidity management and disability prevention (MMDP) interventions for the various sequelae of chronic LF. The recommended minimum package of care includes treatment to kill remaining parasites (adult worms and microfilariae), management of lymphedema and elephantiasis to prevent episodes of acute dermatolymphangioadenitis (ADLA) and surgery for hydrocele [11]. However, the efficacy of each of these elements is understudied and a systematic review and meta-analysis of morbidity management outcomes related to ADLA [12] noted the need for standardisation, high degrees of study heterogeneity and insufficiency of information on the numbers of people requiring MMDP interventions.

Notwithstanding the successes of the GPELF in alleviating the global burden of LF, 45 countries require ongoing preventive chemotherapy and in India alone—where approximately 40% of global infections occur—this amounts to more than 400 million people considered at risk [2]. The urgency to meet the 2030 elimination goals has led to the rapid rollout of IDA as a strategy to accelerate LF elimination in many countries, including India [13]. Since Phase II and III clinical trials of IDA were only completed in 2018 (the latter demonstrating 96% clearance of microfilaraemia after 12 months, far superior to the 32% clearance among participants treated with DA) [5,14], this rollout has been done in conjunction with Phase IV trials that have generated safety and efficacy data from thousands of individuals in India, Southeast Asia and sub-Saharan Africa [6–8].

The rapid rollout of IDA was based, in part, on the findings of transmission dynamics modelling [15]. The models, which used assumptions on the microfilaricidal, macrofilaricidal and sterilizing effects of IDA and DA corresponding to clearance of microfilariae at 12 months after treatment of 100% for IDA and 23.1% for DA, indicated that IDA would have a profound impact on accelerating progress towards the elimination of LF. Yet, community trials have already revealed variable efficacy of the triple drug regimen in Côte d'Ivoire [7], Fiji [16], India [8] and Samoa [17]. Examining the findings of those publications does not allow to fully understand the risk factors associated with poorer treatment outcomes nor to conduct aggregated meta-analyses due to the heterogeneity of study endpoints. Collating, standardizing and analysing the IPD of recent studies would be a productive methodological approach to understand the factors determining variation in treatment responses [18–20].

The positive outcomes of sharing clinical data are well-recognised in the biomedical sciences [21–23], although adherence to data-sharing policies remains variable with the principles of *Findability, Accessibility, Interoperability, and Reusability* often not followed [24]. The need for managed repositories is of critical importance to avoid the unstructured and chaotic

deposition of data over the internet, and the challenge of finding data [25,26]. The Infectious Diseases Data Observatory (IDDO) is an example of an established model based on a repository infrastructure that gathers dispersed and disparate IPD from scattered studies to create standardised datasets that allow researchers to address critical research gaps in a number of infectious diseases, beginning with the Worldwide Antimalarial Resistance Network (WWARN) in 2009 and extending to NTDs, including Chagas disease [27], visceral leishmaniasis [28], schistosomiasis and the soil-transmitted helminthiases [29], and emerging infections including Ebola and SARs-CoV-2 [30]. These platforms focus on the collation, curation and management of information-rich IPD, making them distinct from other population-level epidemiological data repositories such as the Expanded Special Project for Elimination of Neglected Tropical Diseases (ESPEN) portal (https://espen.afro.who.int) and the WHO's Global Health Observatory (https://www.who.int/data/gho).

The first step towards developing a data repository and reuse platform is a feasibility assessment. This entails a systematic literature review to estimate the quantity of IPD potentially available and, critically, the age of such data as a useful proxy of the likelihood of whether they will be available to be shared. Indeed, the fact that the likelihood of data being essentially lost increases with time is further testament to the imperative to archive and safeguard valuable IPD in actively managed repositories [31]. The systematic review also generates valuable study-level meta-data, which can be useful in identifying knowledge gaps in patient characteristics, diagnosis methods, treatment efficacy and safety outcomes [27,28,32–34].

Here, we present the results of a systematic literature review which characterises the published LF treatment studies and clinical trials landscape. We identify studies that have generated pre- and post-intervention IPD on infection and/or morbidity indicators and provide an overview of their essential study-level (meta-data) characteristics. We estimate the abundance of studies and associated IPD that could potentially be available and integrated into a LF data platform and discuss the feasibility and potential usability of such a platform to address knowledge gaps using IPD meta-analyses and identify where further primary research is required.

## Methods

### Systematic review search strategy

We searched the following databases on 5th May 2023: Ovid MEDLINE; Ovid Embase; Ovid Global Health; Scopus; Web of Science Core Collection; World Health Organization Global Index Medicus (AIM (AFRO), IMEMR (EMRO), IMSEAR (SEARO), LILACS (AMRO/OPAS)); Cochrane Database of Systematic Reviews; Cochrane Central Register of Controlled Trials; and Indmed, African Journals Online, ClinicalTrial.gov, WHO ICTRP and Ctri.nic.in. The search strategy was constructed using a comprehensive set of terms representing lymphatic filariasis, including parasite names, disease terms, and appropriate diagnostics. Keywords included (but were not limited to): bancroftian filariasis, brugian filariasis, elephantiasis, *Wuchereria bancrofti*, *Brugia malayi*, *Brugia timori*, lymphedema, circulating filarial antigen. Search strings were tailored for each database and included controlled terminology (e.g., MeSH terms) where relevant. The full search strategy for all databases is given in S1 Text. No limits were placed on language. The search was restricted to those published from 2000 to the date of search (5th May 2023). The review is registered in the international Prospective Register of Systematic Reviews (PROSPERO) under the reference CRD42022319146.

All references were exported to an Endnote 20 library (Thomson Reuters, New York, NY) and deduplicated using a semi-automated method [35,36]. The de-duplicated references were loaded into a Covidence library (www.covidence.org), to facilitate collaborative screening and elimination of studies for the review. References were screened independently by two

members of the review team (any two of LTF, MAK, AU and JBH). We first screened titles and abstracts for eligibility against the inclusion and exclusion criteria. Studies retained at this stage were then evaluated for eligibility by screening the full text.

## Inclusion criteria for published studies on LF treatments

The search strategy was designed to retrieve all published registered trials and cohort studies on treatments for LF (infection and MMDP). For this review, we developed inclusion criteria for a subset of these, namely trials and observational studies published in full-text journal articles since 2000, due to the lower probability of IPD being retrievable and available from studies published prior to this [31].

The inclusion criteria were: (1) published after 2000; (2) availability of full text; (3) participants were diagnosed at baseline with LF by any (microscopy, serology, molecular, ultrasonographic, or clinical) method, and underwent an intervention; (4) a subset of study participants were followed post-intervention for assessment of intervention outcomes. Hence, the inclusion criteria were designed to cover all published studies on LF interventions (drug treatments and MMDP interventions) and which generated individual-level data (IPD) on intervention outcomes.

The exclusion criteria were: (1) non-human animal-only studies; (2) populations not tested for LF indicators; (3) non-primary research studies (books, letters, reviews); (4) case reports/series; (5) qualitative surveys; (6) retrospective studies other than for MMDP interventions; (7) studies published in conference abstracts only, and (8) studies with no follow-ups or follow-ups where individual participants could not have been followed through multiple (at least two) time points, e.g., repeated cross-sections.

## Data extraction and management

A tailored variable data dictionary was prepared to facilitate and standardise the extraction process (S1 Table) through discussion with senior researchers (MGB, AK, MW, AS). A database was created using REDCap (Research Electronic Data Capture) [37] and validated through pilot extraction of variables. Data extraction was undertaken by three researchers (LTF, MAK and AU) and cross-verified among the three researchers and an additional researcher (JBH) was called if any disagreement was identified. The data extracted comprise details on: (1) study settings; (2) design/categories; (3) demographic characteristics; (4) details of drug treatment and MMDP interventions, and (5) an estimate of the number of potential IPD per study arm and at study level. We sourced support for translation of Portuguese publications, but other language publications (of particular note, Chinese) were not within the proficiencies of the review team and hence were not analysed in this review.

## Data analysis

We analysed the study meta-data and characteristics using descriptive statistics, graphics and narrative synthesis. We estimated the amount of potential IPD generated by eligible studies based on reported study arm characteristics. For participants to be included in the estimate, a study arm must have tested or assessed participants for LF indicators (infection or disease), delivered drug treatment or another intervention (to all or a subset of participants) and subsequently followed up (all or a subset of) participants for the collection of post-intervention outcome data (this could include infection or disease indicators). We also made specific estimates on the amount of IPD which could be used to calculate responses to treatment of infection by: microfilaraemia (microfilariae in blood), circulating filarial antigen (CFA), ultrasound to

detect worm nests, filarial dance sign (FDS) or adult worms by estimating the number of participants in those studies who were initially diagnosed by those methods as infection positive.

All analyses were done using Microsoft Excel and R software (version 4.2.2.).

## Results

### Systematic review and screening

A total of 51,193 records were found in the database searches. Through initial deduplication, 33,937 articles were removed. The remaining 17,256 item titles and abstracts were screened. 16,589 irrelevant or further duplicate articles were excluded. From the remaining 667 full-text articles, a total of 147 full-text articles were included which reported on 138 distinct studies (Fig 1 and S2 Text for reference list of all full-text studies). All the data extracted from the 147 full-text articles are included in S2 Table.

### Geographical coverage and estimated abundance of individual participant data

Studies generating longitudinal (at least two time points, one pre- and one post-intervention) IPD on LF interventions (treatments of infection and/or morbidity management interventions) have been carried out in 23 countries (Fig 2). Five of those countries (Brazil, Ghana, Haiti, India and Papua New Guinea) have had multiple studies conducted in each of the past three decades. Five of the 23 countries (Cook Islands, Egypt, Malawi, Sri Lanka and Thailand) have reached the EPHP of LF [2].

### Estimated individual participant data and timelines

We estimate that overall, 29,842 IPD on pre- and post-intervention outcomes have been generated worldwide from studies published between 2000 and 5th May 2023, with Southeast Asia accounting for more than half of the data generated. Between 2000 and 2010, 83 studies were published, which generated an estimated 12,237 IPD and from 2011 to 2022, 55 studies generated 17,605 IPD (Fig 3). Note, however, that a single study undertaken in India and published in 2021 [8] accounts for an estimated 8,803 IPD, i.e., 50% of all IPD generated between 2011 and 2022. We also estimated that, of the 138 eligible published studies, approximately 27% took place prior to 2000, with two studies having taken place as early as 1975. The months of longest follow-up ranged from less than one month to 26 years, with the most frequent longest follow-up time being approximately 12 months after intervention (Fig 4). We identified 13 studies that followed participants for more than 5 years, 2 of those for more than 15 years [40,41].

### Infection and morbidity indicators

Most studies eligible for inclusion (138) have generated IPD on treatments of infection rather than treatments of morbidity (MMDP interventions). Of the estimated 29,842 IPD generated, 14,800 (~50%) comprise pre- and post-intervention (on at least one occasion) microfilaraemia, CFA and/or ultrasound (to detect adult worms) infection indicators, while 6,102 (~20%) comprise pre- and post-intervention morbidity indicators only, i.e., without infection indicators. An additional 300 IPD comprise both infection and morbidity indicators pre- and post-intervention. The remaining 8,940 IPD comprise data where either: (1) the diagnostic measurement was not of microfilariae, CFA, ultrasound or a clinical morbidity indicator; (2) participants were enrolled after the start of the study and thereafter had more than one follow-up (for example, during an observational study / MDA rollout), and/or (3) participants were followed up for adverse events (AEs) but not infection or morbidity indicators.

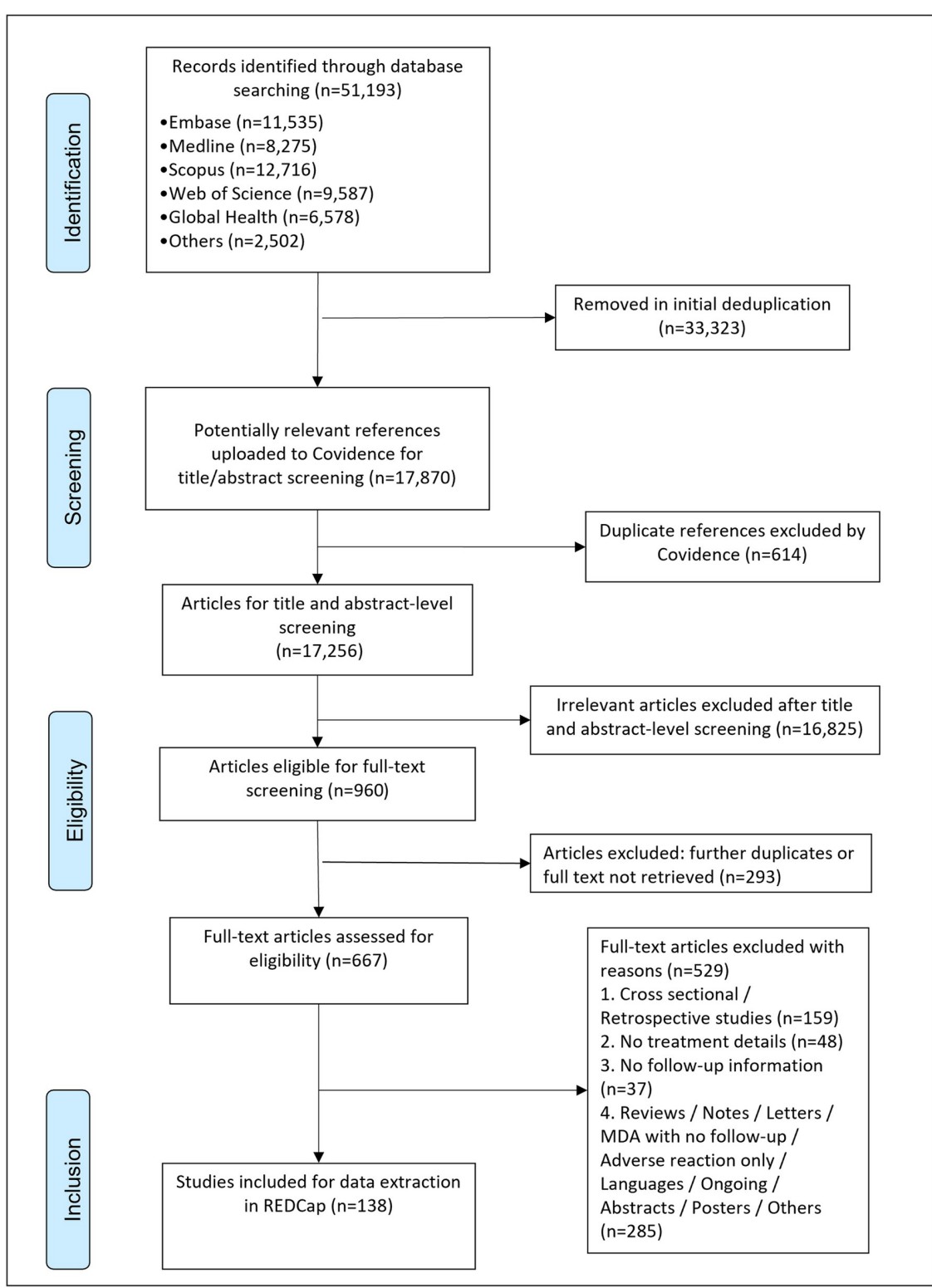

**Fig 1. PRISMA flow diagram of the systematic search and screening.**

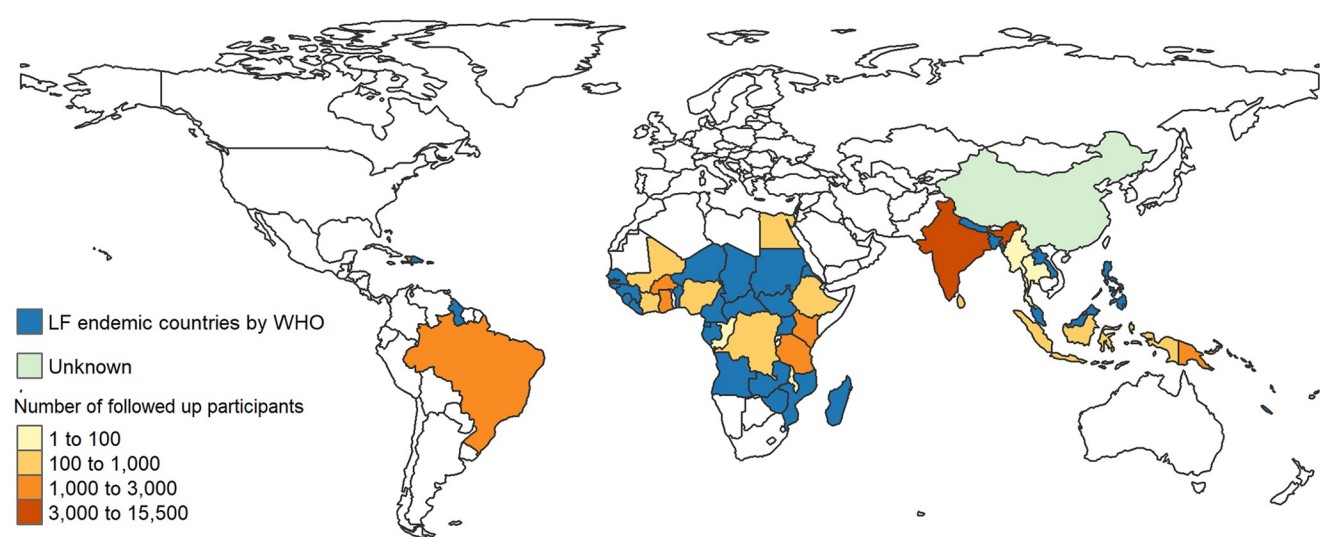

**Fig 2. Geographical coverage of studies collecting individual participant data (IPD) on lymphatic filariasis (LF) infection or morbidity indicators measured pre- and post-intervention.** The 23 countries are shaded from red to yellow in accordance with the estimated abundance of IPD. The 39 countries shaded in blue are endemic for LF but had no studies identified. China is shaded green because of language barriers in assessing the abundance of IPD potentially available. The map was created using World (naturalearthdata.com) and the tmap package [38] for R (v. 4.2.2) [39].

## Diagnostics

We estimate that of the 14,800 IPD on pre- and post-intervention infection indicators, 4,424 comprise only CFA, 1,515 comprise only ultrasound indicators and 2,772 IPD comprise only microfilaraemia. The most common diagnostic used is CFA, with 10,212 IPD generated in

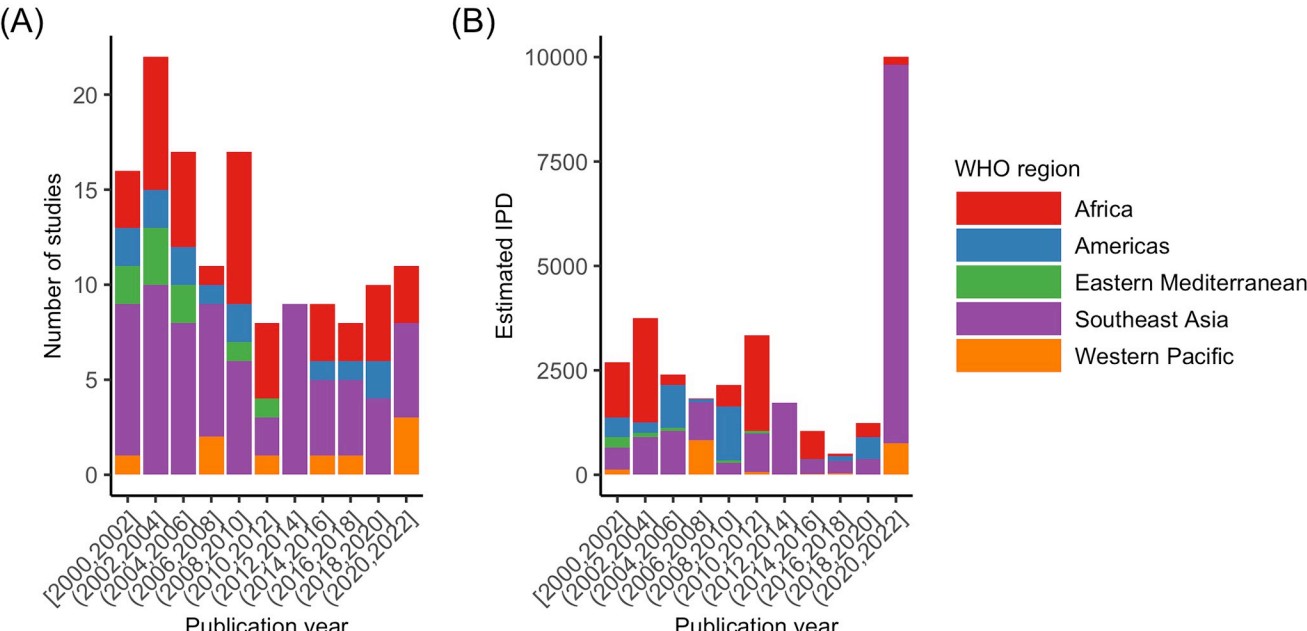

**Fig 3. Frequency distribution (A) and estimated individual participant data (IPD) (B) of 138 eligible studies generating IPD on infection and morbidity indicators measured pre- and post-intervention, grouped by year of study publication.**

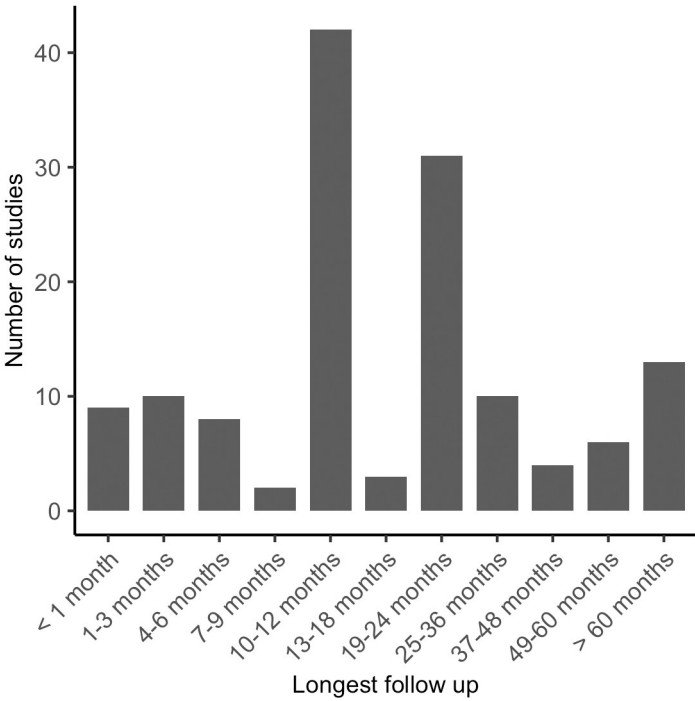

**Fig 4. Frequency distribution of 138 eligible studies generating individual participant data (IPD) on lymphatic filariasis infection and morbidity indicators measured pre- and post-intervention, grouped by months of longest follow-up.**

combination with microfilaraemia and/or ultrasound and/or morbidity indicators. Microfilaraemia was measured in 8,861 IPD in combination with CFA and/or ultrasound indicators and/or morbidity indicators. Note that many IPD contain information from multiple diagnostics, so these estimates do not sum to the 14,800 IPD on pre- and post-intervention indicators. We could not estimate IPD for all specific diagnostic combinations given the information reported in individual studies (Table 1).

## Study design

We identified 116 drug treatment studies. Fifty-six of these are randomized controlled trials, generating 7,918 IPD; 60 are non-randomized, either multi-arm studies or single-arm studies, including observational studies of cohorts followed up after participating in MDA, generating 7,458 potential IPD (Table 2).

**Table 1. Estimated individual participant data (IPD) on lymphatic filariasis infection indicators measured pre- and post-intervention using different diagnostic indicators.**

| Diagnostic | Studies | Estimated IPD[b] |
|---|---|---|
| Microfilaraemia with CFA[a] or ultrasound or morbidity indicators | 91 | 8,861 |
| CFA with microfilaraemia or ultrasound or morbidity indicators | 78 | 10,212 |
| Ultrasound with microfilaraemia or CFA or morbidity indicators | 25 | 2,044 |

[a] circulating filarial antigen

[b] given the information reported in individual studies it was not possible to estimate IPD for all specific combinations (e.g., microfilaraemia with CFA vs. microfilaraemia with ultrasound).

**Table 2. Estimated individual participant data (IPD) on lymphatic filariasis infection indicators measured pre- and post-intervention from different study designs.**

| Design | Studies | Estimated IPD |
|---|---|---|
| Randomized Studies with comparative arms (i.e., with random allocation to interventions) | 56 | 7,918 |
| Non-randomized multi-arm trials; single-arm studies, and observational (including MDA[a]) | 60 | 7,458 |

[a]mass drug administration

**Table 3. Estimated individual participant data (IPD) on lymphatic filariasis infection and morbidity indictors measured pre- and post-intervention from studies reporting sex of trial participants.**

| Sex | Studies | Estimated IPD |
|---|---|---|
| Male & female | 28 | 2,534 & 3,253 |
| Male only | 19 | 2,006 |
| Not reported / unclear | 91 | - |

## Demographic characteristics of study participants

The number of studies and estimated IPD from treatment of infection studies disaggregated by sex and minimum age eligibility are shown in Tables 3 and 4, respectively. These estimates include only the 47 studies where the number of participants by sex can be determined for those who are treated and followed at least once. Of the 91 studies for which these data were not available, some studies reported the demography of the sample population but did not report the age- and sex-structure of the recruited participants. It is noteworthy that 66 of the 109 studies that reported participant age included children <18 years (as well as adults).

## Drug regimens

The most frequently administered drug regimens from 100 of the 116 studies generating pre- and post-intervention infection data and the associated estimate of IPD are shown in Table 5. These regimens account for 11,543 (78%) of the estimated 14,800 IPD from the 116 studies. The most common monotherapy regimens (studies and estimated IPD) are: diethylcarbamazine (47 with IPD 3,206); albendazole (12 with IPD 830); ivermectin (6 with IPD 233); and the anti-*Wolbachia* therapy doxycycline (5 with IPD 99). Corresponding values and estimates for

**Table 4. Estimated individual participant data (IPD) on lymphatic filariasis infection and morbidity indicators measured pre- and post-intervention from studies reporting age eligibility of trial participants.**

| Minimum age eligibility (years) | Studies | Estimated IPD |
|---|---|---|
| < 1[a] | 2 | 427 |
| 1–4 | 16 | 3,489 |
| 5–14 | 35 | 17,502 |
| 15–17 | 13 | 801 |
| ≥18 | 43 | 2,835 |
| Not reported / unclear | 29 | 4,788 |

[a] includes 1 community trial of diethylcarbamazine-fortified salt and 1 trial on participants co-infected with lymphatic filariasis and HIV

**Table 5. Estimated individual participant data (IPD) on lymphatic filariasis infection indicators measured pre- and post-intervention for the most frequently administered drug regimens.**

| Drug | Dose | Duration in days (treatment arms) | Studies (treatment arms) | Estimated IPD |
|---|---|---|---|---|
| Albendazole | 400 mg | 1(12); 7(1) | 12(13) | 793 |
| | 800 mg | 1(1) | 1(1) | 37 |
| Ivermectin | 100 µg/kg | 1(1) | 1(1) | 50 |
| | 150–200 µg/kg | 1(3) | 3(3) | 130 |
| | 400 µg/kg | 1(2) | 2(2) | 53 |
| Diethylcarbamazine | 6 mg/kg | 1(29); 12(8); 14(1); 21(1) | 35(39) | 2,791 |
| | 12 mg/kg | 1(1) | 1(1) | 20 |
| | 25 mg/kg | 90(1) | 1(1) | 13 |
| | 72 mg/kg | 12(1) | 1(1) | 20 |
| | 100 mg | 1(1) | 1(1) | 92 |
| | 300 mg | 1(6); 12(1) | 7(7) | 77 |
| | Unknown | Unknown (1), 12(1) | 2(2) | 193 |
| Doxycycline | 200 mg/day | 42(2); 56(2); 30(1) | 5(5) | 99 |
| Ivermectin + albendazole (IA) | 150–200 µg/kg + 400 mg | 1(14) | 10(14) | 640 |
| | 400 µg/kg + 400 mg | 1(2) | 2(2) | 66 |
| | 400 µg/kg + 600 mg | 1(1) | 1(1) | 15 |
| | 400 µg/kg + 800 mg | 1(3) | 2(3) | 53 |
| Diethylcarbamazine + albendazole (DA) | 6 mg/kg+400 mg | 1(39); 7(4) | 31(43) | 3,265 |
| | 300 mg + 800 mg | 1(1) | 1(1) | 26 |
| | 300 mg + 400 mg | 1(4); 12 (2) | 3(6) | 149 |
| | 100–300 mg (age band)[a] + 400 mg | 1(1) | 1(1) | 864 |
| | Unknown | 2(2) | 2(2) | 119 |
| Ivermectin + diethylcarbamazine (ID) | 400 µg/kg + 6 mg/kg | 1(4) | 2(4) | 100 |
| | 200 µg/kg + 6 mg/kg | 1(2) | 2(2) | 31 |
| Ivermectin + diethylcarbamazine + albendazole (IDA) | 150–200 µg/kg + 6 mg/kg + 400 mg | 1(13) | 10(13) | 1,847 |

[a] age-based fixed dosing 100 mg for 2–5 years and 200 mg for 6–14 and 300 mg for >14 years

combination regimens are: diethylcarbamazine plus albendazole (DA, 37 with IPD 4,423); ivermectin plus albendazole (IA, 15 with IPD 774); ivermectin plus diethylcarbamazine (4 with IPD 131); and ivermectin plus diethylcarbamazine plus albendazole (IDA, 10 with IPD 1,847).

## Morbidity interventions

We identified 33 studies that collected morbidity indicators pre- and post-intervention. Some of these included interventions aimed specifically at treating and managing morbidities, while others examined the effect on morbidity outcomes of treatments primarily for infection. Due to the low anticipated numbers of eligible studies and low numbers of participants, we included retrospective as well as prospective studies for MMDP interventions only, to broaden the exploration of potentially useful IPD. Three of the included studies are retrospective. Eight studies had multiple interventions whose allocation was randomised.

Twenty-eight studies analysed the effect of interventions on limb lymphedema and related acute manifestations. Interventions for limb lymphedema usually include care packages

comprising elements aimed at preventing further damage, and elements promoting lymph flow; some care packages only include one of these categories. There were two studies on the outcomes of surgery for lower limb lymphedema. Eight studies included hydrocele and/or penoscrotal lymphedema (three of which also included limb lymphedema). Outcomes for chronic and acute manifestations were reported heterogeneously.

## Discussion

We have estimated that, from studies published since 2000, approximately 29,842 IPD have been generated from 23 countries on LF infection and morbidity indicators measured before and after intervention. Approximately half of these data are on pre- and post-intervention infection indicators (i.e., treatment of infection studies) following mono-, dual- and triple drug therapies at various doses, covering the full demographic spectrum, from young children to old adults. These studies most frequently use microfilaraemia and/or CFA as outcome measures, with the most common longest follow-up time being 12 months after intervention, but some studies conduct follow-ups after more than 10 years. Approximately 20% of IPD comprises pre- and post-intervention morbidity indicators, revealing a relative paucity of information on the efficacy of morbidity management approaches, despite MMDP being one of the pillars of the LF elimination programme. These data could form the basis of an IPD data repository on the treatment of LF infection and disease.

Since 2000, great progress has been made towards the elimination of LF, with 17 countries having met the criteria for EPHP [2]. These criteria include sustaining infection levels below transmission assessment survey thresholds for at least four years after stopping MDA and providing essential care (MMDP) for known patients. These successes have largely been driven by the WHO's strategy of distributing combinations of antiparasitic medicines to at-risk populations through MDA programmes. Yet, the 2030 target of validating EPHP in 58 (81%) countries [1] has renewed emphasis on the need for optimising treatments for LF to accelerate progress towards this goal. The development and rollout (excluding in areas co-endemic with other filariases in Africa) of triple drug IDA therapy [5,6] has enhanced the feasibility of EPHP within this timeframe, but as with all treatments, responses are variable, and understanding this variability and exploring how to optimise current strategies is needed for sustaining financial and political support to eliminate the disease.

Variation in responses to interventions can relate to geographical factors associated with both host and parasite populations, individual intensities of infection and severity of disease, age and sex differences in bioavailability, method of dosing (e.g., based on weight versus age) and adherence to treatment. Understanding variation requires moving focus from the population to the individual and detailed analyses of IPD, combining data from multiple studies to maximise the power of statistical inference. Traditional meta-analyses of study- or cohort-level data are limited by their inability to incorporate individual-level variables and are also more prone to biases arising from heterogeneity in study designs [42]. Individual participant data meta-analyses provide a solution to this, being able to adjust (statistically) for different study protocols and, therefore, not restricted to completely standardized designs [43]. But such analyses require access to detailed IPD with engagement and collaboration with study investigators to ensure a complete understanding of the original data so that variables and outcomes from multiple studies can be combined and standardized in a reliable and accurate manner. Hence, while the increasingly common sharing of IPD on journal and institutional servers is to be celebrated, it is not sufficient for the most effective data reuse [25,26].

Controlled data repositories provide a more effective means of data sharing, permitting IPD to be stored in one place using consistent standardization with dedicated resources for

curation and collation. Some repositories, such as IDDO, also serve as collaborative hubs for reuse and reanalysis of data and promote fair and equitable data sharing practices that encourage the involvement of data contributors in further analyses [44]. This approach ensures the most robust and reliable scientific results and fosters trust across the scientific community that data will be shared and reused in an equitable and transparent manner. Meta-analyses of IPD hosted by IDDO have been used to inform treatment policies and clinical decision making for malaria [45–47], evaluate the safety of ivermectin in young children [48] and evaluate case definitions for SARS-CoV-2 [49].

The first step in evaluating the feasibility and potential utility of a data platform is performing a systematic review to estimate the abundance of IPD and characterise the landscape of studies that have generated them. Compared to similar exercises conducted for other NTDs—that identified studies conducted over similar timescales, although using somewhat different eligibility criteria—the estimated 29,842 IPD for LF is intermediate to the 20,517 IPD on schistosomiasis [32] and the 35,000 IPD for soil-transmitted helminthiases [33]. Unsurprisingly, most LF IPD have been generated from India, where approximately 40% of worldwide cases occur [2]. More notable is the relative scarcity of IPD from other countries where LF is endemic. For example, fewer IPD have been generated in Indonesia, with a population of approximately 275 million, compared to Papua New Guinea, with a population of around 10 million. There are also numerous countries in sub-Saharan Africa where no IPD have been identified from published studies over the past two decades. Although progress towards elimination in Africa is strong [50], this patchy geographic coverage of IPD could yet prove to be an impediment to further progress.

Geographic heterogeneity explains some of the variability in IPD from different drug regimens. For example, more than 4,000 IPD have been generated on various dose combinations of DA, and nearly 2,000 IPD on IDA, both used for MDA outside of Africa (and particularly in India). Yet for IA (150–200 μg/kg ivermectin + 400 mg albendazole), which remains the only regimen used for MDA within Africa, we could identify only 640 IPD from 14 studies conducted over the past two decades. This is a conspicuously scarce abundance of data for a regimen that is distributed annually to more than 100 million people [2]. This raises concerns about the depth of understanding on the efficacy of this regimen among the individuals comprising the vast and diverse African population. The fact that the 640 IPD come from 14 studies underscores that any future analysis including IA should aim to incorporate the IPD from all of these (small) studies to maximise inferential power.

A notable and distinguishing feature of the studies generating LF IPD is the frequent use of long follow-up times, typically at least one year after intervention. This largely reflects the more complex and protracted effects of current treatment options on filarial parasites, which tend to be more refractory than other nematodes (e.g., soil-transmitted helminths) or trematodes (e.g., schistosomes). For example, in onchocerciasis, while microfilariae are relatively susceptible to treatment, adult filariae macrofilariae often survive, albeit sterilized either temporarily or permanently or with a reduced lifespan [51,52]. The anti-macrofilarial activity of LF treatment options is less well studied, but the protracted decline in CFA (indicative of active macrofilarial infection) compared to the rapid and sustained clearance of microfilariae suggests similar sustained sterilization effects may operate [53]. In principle, the IPD identified here, where both CFA and microfilariae were measured could be combined and, where individuals have also been followed up for multiple years, help disentangle microfilaricidal and anti-macrofilarial activity.

For any data platform, the potential accessibility of IPD is strongly linked with the age of the data, falling by approximately 17% per year [31]. This serves as a reminder that without active engagement and participation with data repositories, valuable data can quickly become

lost to reuse. This is both detrimental to scientific progress and may also be considered as not fulfilling ethical responsibilities to maximise the use of participants' data [54]. Of the most recent (and therefore accessible) data generated since 2016, many are focused on IDA. An IPD meta-analysis of these data could identify individual factors associated with treatment while also accounting for geographical and other study-level heterogeneities (see for example [19] and [20] for similar analyses of responses to treatment of soil-transmitted helminthiasis and schistosomiasis respectively). This could help to explain some of the variation observed in the 12-month efficacies of IDA reported from Phase IV trials which have ranged from 63% in Fiji [16], 71% in Côte d'Ivoire [7] and 84% in India [8], to 94% in Haiti [55], 96% in Papua New Guinea [56] and 96.3% in Indonesia [57]. Indeed, in Fiji, unlike elsewhere, the efficacy of IDA was not superior to DA [16]. Moreover, integration of IPD from ongoing Phase III clinical trials of moxidectin [58] given with combinations of albendazole and diethylcarbamazine (https://clinicaltrials.gov/study/NCT04410406) within such meta-analyses will also be important in generating comprehensive assessments of the relative efficacies of the different options available for treating LF infection.

While our quantification of IPD here has focused on pre- and post-intervention infection and morbidity indicators (i.e., longitudinal data measured before and at least once after intervention), these data will also frequently include information on safety and tolerability (although we did not quantify this explicitly because safety data typically do not require the longer follow-up of individuals that is required to assess infection or morbidity responses to intervention). Adverse events associated with the treatment of LF are common, albeit usually transient and seldom severe, and often relate to the killing of microfilariae [59–61]. This is common among filarial nematodes and indeed the severity of AEs induced by the rapid killing of *Onchocerca volvulus* and *Loa loa* microfilariae following treatment with diethylcarbamazine led to its withdrawal from use in Africa [62], although its re-introduction as part of IDA has been successfully trialled in areas non-endemic for onchocerciasis and loiasis [7], and strategies for its potential wider use in Africa were discussed [63]. It is routine for studies to report AEs at a cohort level, sometimes with more detailed individual analyses or clinical investigation of more serious events. But with abundant IPD, one could determine comprehensively whether the probability of AEs is associated with individual-level factors [59]. For example, a threshold level of *L. loa* microfilariae above which treatment with ivermectin (which also kills microfilariae) is contraindicated has been used during Test-and-(not) treat pilot field trials [64] because of an unacceptably high probability of severe AEs. Such a quantification could be extremely useful if DEC is to be reintroduced into Africa as part of IDA.

The efficacy of MMDP interventions has only been sparsely examined in published literature (e.g., [65]). The studies identified in this review where interventions for morbidity management were carried out were diverse with highly heterogeneous reporting of study-level meta-data and outcomes. Additionally, studies testing interventions for management of lymphedema or hydrocele may mix participants with both filarial and non-filarial causes (these studies were not included in this review). This explains why, hitherto, meta-analyses of morbidity management interventions for LF [12] have been restricted to only a small number of studies and on a subset of interventions and outcomes for particular complications. Further and more standardised studies are urgently needed in this domain and should include post-surgery follow-up of participants to quantify relapse rates and the occurrence of secondary infections. Additionally, as with data on infection, analysis of individual-level data would be especially valuable for understanding the drivers of variation in responses to treatment, and pooling (harmonised) data from individual studies would increase statistical power where subgroups of interest within individual studies are small.

The principal goal of this work was to identify studies and estimate the abundance of IPD on LF infection and morbidity indicators measured before and after intervention, which could potentially be integrated into a global data repository. Although we took a conservative approach to the estimation of IPD (e.g., counting only participants who were successfully followed at the last follow-up time), our estimates are based on information reported in publications and will be inexact. For example, it was often difficult to determine from the reported information whether the same cohort of individuals had participated before and after intervention, and in several studies additional participants were recruited after the initial intervention. It was also a challenge to avoid 'double counting' of IPD generated from single studies but reported in multiple publications and the often-limited reporting of study design made classification difficult. These challenges will naturally introduce uncertainty in our estimates of IPD. The difficulty in estimation using reported information is particularly apparent when trying to quantify IPD associated with individual-level variables. For example, most studies provided only age eligibility criteria of participants, so we were not able to estimate a detailed demographic breakdown of IPD. Similarly, for sex, only 47 of the 138 studies reported sufficient information to estimate IPD for males and females to reach conclusions on gender representativeness, although notably, we identified 19 studies that included male only participants and no female-only studies.

## Conclusions

This review has highlighted a substantial number of recent studies on LF infection and morbidity responses to interventions, which if IPD were available, could be highly valuable in improving understanding of the factors that shape variability in responses to treatments. This is the first step in building a case for the utility and feasibility of an IPD data sharing and reuse platform that could maximise the power of these data. The next stage of this process will be to enhance engagement with the LF research community, seeking their commitment to effective data sharing and their expertise in defining and prioritising research questions that could be usefully answered using an operational platform.

## Supporting information

**S1 Text. Database search strategies.**
(DOCX)

**S2 Text. Reference list of 147 full text articles reporting on 138 eligible studies.**
(DOCX)

**S1 Table. Variable data dictionary.**
(XLSX)

**S2 Table. Complete data extracted from 147 full text articles on 138 eligible studies.**
(XLSX)

## Author Contributions

**Conceptualization:** Julia B. Halder, Sauman Singh-Phulgenda, Philippe J. Guérin, Maria-Gloria Basáñez, Martin Walker, Adinarayanan Srividya.

**Data curation:** Luzia T. Freitas, Mashroor Ahmad Khan, Azhar Uddin, Julia B. Halder.

**Formal analysis:** Luzia T. Freitas, Mashroor Ahmad Khan, Azhar Uddin, Julia B. Halder, Martin Walker.

**Funding acquisition:** Sauman Singh-Phulgenda, Manju Rahi, Philippe J. Guérin, Maria-- Gloria Basáñez, Ashwani Kumar, Martin Walker, Adinarayanan Srividya.

**Investigation:** Luzia T. Freitas, Mashroor Ahmad Khan, Azhar Uddin, Julia B. Halder, Jeyapal Dinesh Raja, Vijayakumar Balakrishnan, Adinarayanan Srividya.

**Methodology:** Luzia T. Freitas, Julia B. Halder, Sauman Singh-Phulgenda, Eli Harriss.

**Project administration:** Julia B. Halder, Sauman Singh-Phulgenda.

**Resources:** Luzia T. Freitas, Mashroor Ahmad Khan, Azhar Uddin, Julia B. Halder.

**Supervision:** Luzia T. Freitas, Julia B. Halder, Sauman Singh-Phulgenda, Maria-Gloria Basáñez, Martin Walker, Adinarayanan Srividya.

**Visualization:** Luzia T. Freitas, Mashroor Ahmad Khan, Azhar Uddin, Julia B. Halder, Martin Walker.

**Writing – original draft:** Martin Walker.

**Writing – review & editing:** Luzia T. Freitas, Mashroor Ahmad Khan, Azhar Uddin, Julia B. Halder, Sauman Singh-Phulgenda, Jeyapal Dinesh Raja, Manju Rahi, Matthew Brack, Philippe J. Guérin, Maria-Gloria Basáñez, Ashwani Kumar, Martin Walker, Adinarayanan Srividya.

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
