## [Decision Letter · Decision Letter 0]

24 Oct 2023

Dear Dr Walker,

Thank you very much for submitting your manuscript "The lymphatic filariasis treatment study landscape: a systematic review of study characteristics and the case for an individual participant data platform" for consideration at PLOS Neglected Tropical Diseases. As with all papers reviewed by the journal, your manuscript was reviewed by members of the editorial board and by several independent reviewers. The reviewers appreciated the attention to an important topic. Based on the reviews, we are likely to accept this manuscript for publication, providing that you modify the manuscript according to the review recommendations. 

Sincerely,

Sabine Specht

Academic Editor

Francesca Tamarozzi

Section Editor

Reviewer's Responses to Questions

**Key Review Criteria Required for Acceptance?**

**Methods**

-Are the objectives of the study clearly articulated with a clear testable hypothesis stated?

-Is the study design appropriate to address the stated objectives?

-Is the population clearly described and appropriate for the hypothesis being tested?

-Is the sample size sufficient to ensure adequate power to address the hypothesis being tested?

-Were correct statistical analysis used to support conclusions?

-Are there concerns about ethical or regulatory requirements being met?

Reviewer #1: The methods were clearly stated from the outset and were appropriate to achieve the stated aim of the review. The initial collection was clearly very extensive as shown by the very large attrition rate when removing duplicates from the large number of publication depositories. Overall no concerns

Reviewer #2: The authors have done a systematic review of literature to estimate the abundance of Individual participant data(IPD)on pre- and post-intervention indicators of LF infection and/or LF morbidity and assess the feasibility of building a global data repository. This is also aimed at developing a better strategy for elimination of LF. The study design is appropriate but the real data the authors could make available is small compared to the abundance of literature.Due to problems with the research studies itself the authors could select only 147 full text articles for analysis. I would like to congratulate the authors for the commendable and systematic efforts to identify he eligible articles.There are no ethical or regulatory concerns . The statistical methods used are appropriate

**Results**

-Does the analysis presented match the analysis plan?

-Are the results clearly and completely presented?

-Are the figures (Tables, Images) of sufficient quality for clarity?

Reviewer #1: The results are clearly and appropriately presented. The figures and tables are clear and provide additional information that assists in the understanding of the results.

Reviewer #2: The analysis is presented as per the plan itselfThe authors were looking for the IPD regarding management of infection (preventive chemotherapy) and that of management of disease, the morbidity management and disability prevention. The tables and figures explains well the results of analysis. The results are well presented

**Conclusions**

-Are the conclusions supported by the data presented?

-Are the limitations of analysis clearly described?

-Do the authors discuss how these data can be helpful to advance our understanding of the topic under study?

-Is public health relevance addressed?

Reviewer #1: The conclusions are clear and the limitations are adequately discussed. Since the main aim of this work was to identify material currently available to advance public health interventions in LF, these are covered in detail. Overall, this paper provides the benchmark for future work collection and curating individual patient data in LF and to plan prospective data collection. Overall, the aims of the study and publication of the material are achieved.

Reviewer #2: The authors had limitations to obtain the IPD on morbidity management and disability prevention. The Global program to eliminate LF and its preventive chemotherapy is a well designed public health program and lot of data have been already generated on that. But the other strategy of MMDP has not been taken up by all the countries and the data that could be made available in this study is minimal only. This is the limitation of tis study also

**Editorial and Data Presentation Modifications?**

Reviewer #1: The paper is well written and easily understandable. it should be accepted without the need for additional modification.

Reviewer #2: Here the goal of the authors is to develop a global data repository for LF infection and LF morbidity. For this the authors have resorted to doing the systematic eview of the available studies. The massive data available on LF elimination data- the preventive chemotherapy is the Global health observatory for lymphatic filariasis. Here the data of the number of ppeople to whom preventive chemotherapy was given, percentage consumption in the country and lal details are available. But authors have not mentioned about this at all. As a reviewer I would like to have an opinion from the authors who have done lot of work, on how this repository will help them or not help them to achieve their goals of this study. Data on MMDP also will be available but may not be full proof. 

The authors may be encouraged to get details of global health observatory LF and may be added here in this manuscript how that would help to develop a repository important for LF elimination

I am not in a position to give recommendation but once this is done this can be considered for acceptance

**Summary and General Comments**

Reviewer #1: The output of this work has considerable importance and has the potential to answer many of the questions that have dogged the LF community. The identification of areas where data is lacking, for example in MMDP studies and in treatment data from Africa, is an important step, although those in the field would be able to identify these from their own experience. The key step is to identify what data is often missing from data sets and ensure that these are collected in the future. The need to develop guidelines for collection and presentation is essential and probably reaches across all studies and diseases. It came as a surprise that even simple things like age and sex are an issue in these data sets.

It is impressive that the data cut off for analysis was as late as May this year, meaning that an enormous amount of work in writing has been achieved in a very short space of time.

Reviewer #2: All the authors have done lot of committed work from the designing, collection of data, analysis and writing the manuscript. Similarly lot of efforts have gone into writing this manuscript also. So this is to be considered as a good write up in this context which could become an inspiration to other researchersSo with revisions this may be accepted. The authors have also explained the limitations of the study

PLOS authors have the option to publish the peer review history of their article (what does this mean?). If published, this will include your full peer review and any attached files.

Reviewer #1: Yes: John Horton

Reviewer #2: No

Figure Files:

Data Requirements:

Reproducibility:

References

---

## [Editor Report · Decision Letter 1]

22 Dec 2023

Dear Dr Walker,

We are pleased to inform you that your manuscript 'The lymphatic filariasis treatment study landscape: a systematic review of study characteristics and the case for an individual participant data platform' has been provisionally accepted for publication in PLOS Neglected Tropical Diseases.

Best regards,

Sabine Specht

Academic Editor

Francesca Tamarozzi

Section Editor

---

## [Editor Report · Acceptance letter]

10 Jan 2024

Dear Dr Walker,

We are delighted to inform you that your manuscript, "The lymphatic filariasis treatment study landscape: a systematic review of study characteristics and the case for an individual participant data platform," has been formally accepted for publication in PLOS Neglected Tropical Diseases.

Best regards,

Shaden Kamhawi

co-Editor-in-Chief

Paul Brindley

co-Editor-in-Chief
